# Achieving Antimicrobial Stewardship on the Global Scale: Challenges and Opportunities

**DOI:** 10.3390/microorganisms10081599

**Published:** 2022-08-08

**Authors:** Jorge Pinto Ferreira, Daniela Battaglia, Alejandro Dorado García, KimAnh Tempelman, Carmen Bullon, Nelea Motriuc, Mark Caudell, Sarah Cahill, Junxia Song, Jeffrey LeJeune

**Affiliations:** Food and Agriculture Organization of the United Nations (FAO), 00153 Rome, Italy

**Keywords:** antimicrobial resistance (AMR), antimicrobial use (AMU), sustainable development goals (SDGs), antimicrobial stewardship (AMS), Codex, InFARM, governance, FAO, food safety, animal health, livestock production

## Abstract

Antimicrobial resistance (AMR) has been clearly identified as a major global health challenge. It is a leading cause of human deaths and also has a toll on animals, plants, and the environment. Despite the considerable socio-economic impacts, the level of awareness of the problem remains woefully inadequate, and antimicrobials are not generally recognized as a global common good, one that everyone has a role and responsibility to conserve. It is imperative for antimicrobial stewardship to be more widely implemented to achieve better control of the AMR phenomenon. The Food and Agriculture Organization (FAO) of the United Nations plays an important role in promoting and facilitating antimicrobial stewardship. The specific needs to be addressed and barriers to be overcome, in particular, in low- and middle-income countries in order to implement antimicrobial stewardship practices in agrifood systems are being identified. As a global community, it is essential that we now move beyond discussing the AMR problem and focus on implementing solutions. Thus, FAO provides multi-pronged support for nations to improve antimicrobial stewardship through programs to strengthen governance, increase awareness, develop and enhance AMR surveillance, and implement best practices related to antimicrobial resistance in agrifood systems. For example, FAO is developing a platform to collect data on AMR in animals and antimicrobial use (AMU) in plants (InFARM), working on a campaign to reduce the need to use antimicrobials, studying the use of alternatives to the use of antimicrobials (especially those used for growth promotion) and actively promoting the implementation of the Codex Alimentarius AMR standards. Together, these will contribute to the control of AMR and also bring us closer to the achievement of multiple sustainable development goals.

## 1. Introduction

Antimicrobials are naturally occurring semi-synthetic or synthetic substances that kill or inhibit the replication of microorganisms. Remarkable achievements have been reached in human medicine due to the availability and efficacy of antimicrobials, from the treatment of previously fatal infections to surgical procedures, otherwise impossible. In parallel, the use of antimicrobials in animals (terrestrial and aquatic) and plant production and protection have also contributed to increased animal health and welfare, agricultural outputs, and, overall, food safety and security and the economic wellbeing of millions of agricultural producers and livestock keepers across the globe. Antimicrobial resistance (AMR) can be defined as the inherited or acquired characteristic of microorganisms to survive or proliferate in concentrations of an antimicrobial that would otherwise kill or inhibit them. In practical terms, the view of AMR is often simplistic and narrow, referring only to the failure of antibiotics to treat human patients. The true dimensions of this problem are, however, much broader. Although all antibiotics are antimicrobials, not all antimicrobials are antibiotics. The term antimicrobial encompasses a much larger group of agents with diverse targets, including some heavy metals, some antiparasitic drugs, fungicides, and biocides. Moreover, antimicrobials are not only used in humans but also in many aquatic and terrestrial animals (to treat, prevent and control diseases and even for growth promotion) and crop/plant production and protection. Many of the same antimicrobial classes are used in humans and animals, and a few of these are also used in plants. The transfer of resistance genes among bacteria present in humans, animals, plants, and the environment underscores the need for antimicrobial stewardship in all domains, including in human health, agrifood systems, and the environment. The common use of antimicrobials in animal and plant production systems has raised concerns for the individuals using and applying these compounds in agrifood production systems, the safety of food, and environmental contamination with both antimicrobials and antimicrobial-resistant microorganisms, which are challenging to quantify and trace, and complicate mitigation strategies. These concerns are not limited to the use of antibiotics but also the use of some heavy metals and fungicides. For example, the use of triazole fungicides for protecting crops has contributed to anti-fungal resistance in the pathogen responsible for the human disease aspergillosis [1].

Antimicrobial resistance threatens the attainment of many of the 2030 Sustainable Development Goals (SDGs). Recent modeling work on the global impact of AMR estimated that antimicrobial-resistant infections directly caused 1.27 million deaths in 2019 and played an indirect role (e.g., failures in surgery) in 4.95 million deaths, placing AMR as a leading cause of human deaths worldwide [2]. Also important are the impacts on animal health, welfare, and productivity, which, along with threats to plant health, threaten global food security, the overall economic sustainability of the agrifood sector, and the economic development associated with trade in agrifood products.

The solution to AMR will unfortunately not be the development of new antimicrobials. Very few novel antimicrobial classes have been discovered and commercialized since the late 1980s. The process of drug development and approval is long and resource intensive. Moreover, resistance is also predicted to develop against any newly developed antimicrobial. Thus, the need to preserve the efficacy of the classes of antimicrobial that are currently available is imperative.

Because AMR knows no species, sector, or geographic borders, and some antimicrobials are used across humans, animals, and plants/crops, it represents a perfect example of a challenge that could benefit greatly from a One Health approach on a global scale. The definition of “One Health”, according to the One Health High-Level Expert Panel (OHHLEP), is “an integrated, unifying approach that aims to sustainably balance and optimize the health of people, animals, and ecosystems. It recognizes that the health of humans, domestic and wild animals, plants, and the wider environment (including ecosystems) are closely linked and inter-dependent. The approach mobilizes multiple sectors, disciplines, and communities at varying levels of society to work together to foster well-being and tackle threats to health and ecosystems while addressing the collective need for clean water, energy, and air, safe and nutritious food, taking action on climate change, and contributing to sustainable development”.

In 2015, a global action plan on AMR was among the first attempts to coordinate global action on AMR [3]. Subsequently, multiple additional resolutions and strategies have been adopted by international organizations, such as the World Health Organization (WHO), the World Organization for Animal Health (WOAH, formally known as OIE), and the Food and Agriculture Organization of the United Nations (FAO). More recently, the United Nations Environment Program (UNEP) formally joined these efforts against AMR under the Quadripartite agreement (see Section 2.2.1). Coordinating and implementing the One Health concept can be challenging, as the motivations and interests of the related different stakeholders can be different, but this quadripartite agreement provides a strong global signal of the path that needs to be taken.

The purpose of this paper is to describe some of the main initiatives that FAO has engaged in to promote and enable antimicrobial stewardship (AMS), as outlined in the FAO Action Plan on AMR (2021–2025) [4], including initiatives specifically targeted for the plant production sector. It highlights some of the fundamental challenges and opportunities and presents successful case studies that, once adapted to the different local realities, can collectively contribute to further control AMR. Increased awareness of these initiatives and the promotion of engagement therein are the ultimate goals of this review.

## 2. Antimicrobial Stewardship: Basis of a Global Framework

Promoting and supporting antimicrobial stewardship (AMS) at the global, national, and local levels is dependent on building strong supporting pillars. From the FAO perspective, these pillars are (1) awareness, (2) governance, (3) practices, and (4) surveillance. The following looks at some of the key elements of these pillars and some of the tools and resources available to build them (Figure 1).

### 2.1. Awareness

Despite the alarming numbers of human deaths, the awareness of the AMR phenomenon is still limited for most lay people. The terminology used in this field can be frequently misinterpreted, with a common confusion between “the body” being resistant vs. the actual microorganism being resistant to the antimicrobial used [5]. Even in medical fields, AMR is frequently not assumed as the cause of death, but instead the disease (e.g., pneumonia, meningitis, urinary tract infection) that was not possible to successfully treat. Risk perception is disproportionate to the actual (real) risk. Such risk discounting has the advantage of reducing fear and panic but, at the same time, leads to apathy and failure to act. Given the lack of awareness of the value of antimicrobials in maintaining public and animal health, and food security, many lay citizens may fail to recognize that antimicrobials are common goods [6], and, because of the scale, magnitude, severity, and multiple drivers of resistance, everyone’s actions, including individual behavior (e.g., proper disposal of unused antimicrobials), can and must contribute to combating AMR and preserving the efficacy of these important medicines.

From an agrifood perspective on the global scale, the most recent FAO Action Plan on AMR (2021–2025) [4] highlights the need to increase stakeholder awareness and engagement. Among the key objectives are to:Leverage participatory approaches to better understand stakeholder perspectives and motivations;Identify barriers to change and pilot collaborative solutions for a science-based approach to interventions; andEnable, empower, and incentivize stakeholders to transform knowledge and awareness of AMR risks into action.

The following sub-sections look globally at FAO efforts to raise awareness and examine barriers to change and participatory actions that can help identify and overcome them.

#### 2.1.1. Effective Awareness Raising—Global Level

To more effectively raise awareness and transform it into action, the objective and target audience for awareness raising must be clear. At a global level, FAO leverages and synergizes its organizational strengths by partnering with other organizations and entities to keep AMR on the global agenda in an ongoing effort to secure ongoing high-level commitment to address this challenge. An example of one recent initiative, in collaboration with Johns Hopkins University/React, introduces AMR/AMU into strategic policy venues where these issues might be taken up for discussion by key stakeholder groups, from government officials to civil society.

Providing access to knowledge is also critical so that those advising policymakers and politicians have up-to-date information that will inform the brief of those partaking in international dialogues. In addition to publishing data, FAO started a monthly Knowledge Dissemination Dialogue webinar series as a means of making experts and knowledge more accessible to all countries. These data-rich webinars bring participants up to date on specific scientific and technical topics related to AMR. These may include, among others, microbiology, epidemiology, environmental or behavioral science, plant and animal production, and health. They are open, free of charge, and available on the FAO YouTube channel [7] and provide an opportunity for everyone, everywhere, to learn about cutting-edge approaches to address AMR from some of the world’s most respected “thinkers and doers”.

Social media and communication channels that can be widely and easily accessed through cell phone technology are key in order to reach wider audiences. Thus providing context-specific information on AMR and simple messages and actions for different sectors through different social media channels are increasingly gaining traction.

To increase awareness raising, communication, and advocacy among the plant production and protection sectors, the following initiatives are ongoing:Strengthening the FAO AMR website as it relates to plant health;Developing factsheets, policy briefs, and guidance documents on the use of antibiotics and fungicides in plant agriculture;Organizing information webinars and workshops for FAO member countries on AMR and AMU in the plant health sector to improve farmer’s knowledge of AMR and AMU in plant production and protection and on alternatives to minimize the use of pesticides that contain antimicrobial substances by reviewing extension material and implementing Farmer Field School Programs focusing on, for example, integrated pest management (IPM), biological control and the use of disease-resistant crop varieties. These webinars and workshops are good opportunities to clarify, for example, that in some countries, antibiotics are classified as pesticides, and farmers might be using them without being properly aware of this fact;Improving farmer knowledge on AMR and AMU in plant production and protection and on alternatives to minimize the use of pesticides that contain antimicrobial substances by reviewing extension material and implementing Farmer Field School Programs focusing on, for example, integrated pest management (IPM), biological control and the use of disease-resistant crop varieties.

The related work plan will be implemented through a coordinated, multisectoral, One Health approach that is key to addressing AMR in the context of the 2030 Agenda for Sustainable Development and in support of the FAO Action Plan on AMR 2021–2025.

#### 2.1.2. Barriers to Change—Local Level

AMR can be considered a social problem, thereby requiring social solutions [8]. Consequently, theory and methods from across the social sciences are necessary to undercover and understand the multiple and interacting environments and drivers from which AMR emerges and is transmitted among people, animals, plants, and the environment. These environments and factors are influenced and impacted by the activities of those manufacturing, regulating, selling, prescribing, and using antimicrobials, with many practices along value chains being enabled, encouraged, or constrained by broader sociocultural, political, and economic structures. Benchmarking tools (farmer-to-farmer comparison) can also have a potential role in enabling livestock producers to see how their AMU compares to comparators in the national and production system context, and therefore they can represent a relevant approach to engaging livestock producers in changing behaviors.

To begin understanding these influencers and drivers, the FAO has been exploiting the power of mixed-methods qualitative and quantitative approaches to identify the knowledge, attitudes, and practices that drive AMR, including patterns of antimicrobial use and practices for on-farm infection prevention and control. This “bottom-up approach” emphasizes that interventions and strategies to address AMR must be evidence-based and should draw upon knowledge, attitudes, and practices displayed by a wide range of stakeholders with the power to affect AMR in the agrifood sector, including farmers, animal health professionals, agrifood producers, policymakers and citizens [9].

The benefits to AMR of this bottom-up approach are exemplified by an international study supported by the Fleming Fund of the United Kingdom. Broadly, this study assessed patterns of knowledge, attitudes, and practices among users, sellers, and prescribers of antimicrobials within livestock systems across five sub-Saharan African countries, including Ghana, Kenya, Tanzania, Zambia, and Zimbabwe [9]. On a typical mission, the FAO research team, working alongside government officers, would conduct focus groups and in-depth interviews among farmers, agrovets (sellers of veterinary medicines), and animal health professionals from the public and private sectors. Qualitative data from these interviews would then be examined for reoccurring themes and inform the development of quantitative knowledge, attitudes, and practices survey. To administer the survey, enumerators from local communities would be trained in ethnographic methods, and thereafter, the survey would be administered to around 200 farmers. Community engagement is critical. After analyzing the survey data for risk factors related to the emergence and spread of AMR, the research team would return to the community to discuss results, with a focus on further understanding the factors driving risky practices and the barriers to the adoption of best practices.

Results from the studies such as the ones described above highlight several important challenges to addressing AMR within low- and middle-income countries and in the animal health sector. First, across countries and livestock production systems, a person’s knowledge and attitude toward AMR did not predict their AMR-relevant practices [9,10]. That is, those who recognized that AMR was a problem driven by inappropriate use of antimicrobials were no more likely to use antimicrobials prudently. This knowledge-practice gap stresses that, alone, awareness-raising campaigns, which to date have played a major role in global and national strategies to address AMR (e.g., World Antimicrobial Awareness Week), are unlikely to motivate better practices and thus ultimately have limited effect on reducing AMR [11]. Awareness campaigns need to be targeted with clearly associated theories of change and expected outcomes that can be measured to determine reasons for suboptimal success. A second major barrier to addressing AMR in agrifood systems in the countries studied was the limited influence of animal health professionals in the treatment of livestock [9,12,13]. The professional animal health sector in many low- and middle-income countries, due to long-term divestment in public services since the 1980s and 1990s, does not always have the capacity to ensure sick animals receive the proper diagnosis and proper treatment. Consequently, even if farmers believed animal health professionals should always be consulted when their animals were sick, a lack of access to this source of reliable information limited the translation of these attitudes into action. Together, these observations highlight the importance of structural factors (e.g., historical, economic, and political) in patterning AMR.

To address these barriers, the FAO has begun implementing AMR Farmer Field School (FFS) programs in selected countries. In a typical FFS, a group of farmers (15–20) meets on a weekly basis at a demonstration farm over the course of a production cycle (e.g., 6 weeks for broiler poultry). The school proceeds under the guidance of a trained FFS facilitator, who is often a local animal health or agricultural extension worker. FFS are well suited to addressing the knowledge-practice gap as the approach is based on the principles of adult-centered learning (e.g., “learning-by-doing”) and the exchange and creation of knowledge through group-level problem solving. In addition, while the schools themselves cannot address resource constraints in the animal health sector, they can provide an opportunity for farmers to work with their local animal health professionals to develop trust and thus maximize the capacities available. A recent evaluation study comparing participants in layer poultry farming FFS in Kenya and Ghana to non-FFS participants shows that FFS participants reported more prudent practices, including reducing and even eliminating the use of antimicrobials in layer production. In addition, FFS participants reported greater investments in biosecurity and were more likely to call an animal health professional upon the first sign of disease in their flocks [14].

### 2.2. Governance

Governance is an essential component of global, national, regional, and local responses to AMR. In this section, “AMR governance” refers to the diffuse and collaborative processes and systems through which a range of public and private actors and entities articulate their interests, frame and prioritize issues, and make, implement, monitor, and enforce AMR-related decisions. There is a broad consensus that AMR governance must adopt a One Health approach [15]. Because AMR has multiple drivers that should be tackled on many fronts, a One Health approach supports the creation of policy, legal and institutional frameworks for AMR that are holistic, integrative, and coherent [16]. Importantly, all activities should fully integrate plant production and environmental representation, two sectors that have been typically underrepresented in AMR discussions. Effective governance guides the sustainable management of AMR, and this depends on political will and a well-informed institutional framework for innovating, evaluating, and strengthening policies and legislation. Strengthening AMR governance is another one of the objectives of the FAO Action Plan on AMR 2021–2025 [4].

#### 2.2.1. Global AMR Governance

The United Nations General Assembly (UNGA) Resolutions in December 2015 [17] and September 2016 [18] recognized the magnitude of AMR as a global threat. The latter resolution contains the Political Declaration of the High-level Meeting on Antimicrobial Resistance, which served as the basis for the establishment of the ad hoc Interagency Coordination Group (IACG) on Antimicrobial Resistance [18]. The report of the IACG reinforces the role of AMR governance, both at the global and the national level. In furtherance of IACG’s mandate for improved coordination, recommendations were made to strengthen accountability and global governance by establishing three inter-related structures [19]: (1) An Independent Panel on Evidence for Action against Antimicrobial Resistance in a One Health context (Independent Panel, IPEA); (2) A One Health Global Leaders Group (GLG) on Antimicrobial Resistance, supported by a Joint Secretariat managed by the Tripartite Organizations (FAO, WOAH, and WHO); and (3) A multi-stakeholder partnership platform (MPP) to facilitate multi-stakeholder engagement on AMR. The Independent Panel aims to inform the agenda of the GLG. The latter’s discussions are also used to drive the agenda of the MPP and the issues resulting from MPP discussions subsequently feed into the work of the GLG and the Independent Panel [19].

In 2018, FAO, WOAH, and WHO signed a Memorandum of Understanding (MoU) regarding cooperation to combat health risks at the animal/human/ecosystems interface, in the context of the “One Health” approach and including antimicrobial resistance. In 2022, UNEP signed the MoU, making it quadripartite. The new quadripartite MoU provides a legal and formal framework for the four organizations to tackle the challenges at the human, animal, plant, and ecosystem interface using a more integrated and coordinated approach. The quadripartite has, among other initiatives, developed a joint strategy on AMR, which includes as part of its outcomes the support to countries to develop policy and legislative effective and country-owned AMR responses.

#### 2.2.2. AMR Governance at the National Level

Two key dimensions of AMR governance at the national level are discussed here. The first involves the coordination of AMR national policies through a multidisciplinary entity. The second entails the strengthening of regulatory (policy, legal and institutional) frameworks that are relevant for AMR. Many national action plans (NAPs) [20] display both these dimensions, in addition to the identification of governance and regulatory frameworks as specific objectives.

A national-level coordinating mechanism must bring together all stakeholders, including government ministries, parliamentarians, civil society organizations, the private sector, academia, and regional and international partners. Such a coordinating body should have a clear mandate or terms of reference (which may include overseeing the implementation of the AMR policy), an established budget, a defined accountability framework, and should integrate actions and interventions vertically within a sector or horizontally across sectors [21]. According to the FAO AMRLEX [22], the data set of national and supranational legislation set up by FAO, only a few countries have formally established coordinating mechanisms for AMR governance by law. In the absence of a legal foundation, the coordinating entity would depend solely on fluctuating political will; accountability is reduced, implementation may be inhibited, and effectiveness and sustainability could prove challenging [23].

The second element of AMR governance at the national level involves the alignment of sector-specific national policies and legislation with relevant international standards [24], such as those of the Codex Alimentarius, applicable to the sector as well as international best practices to prevent and control AMR. With the purpose of guiding countries in the analysis of their legal frameworks, FAO developed a Methodology to analyze AMR-relevant legislation in the food and agriculture sectors (2015) [25]. This methodology was applied in 27 countries across different continents and also by three regional organizations, with financing from the Multi-Partner Trust Fund (MPTF) for AMR, FAO, WOAH, and WHO developed a One Health Legislative Assessment Tool, which is designed to replace the methodology, to guide country-level assessment of the weaknesses and gaps in legislation across sectors relevant for AMR. The outcomes of the assessments can then be used to inform legislative reform as a next step. The tool comprises a cross-cutting chapter on AMR governance, as well as sectoral chapters on human health, food safety, veterinary legislation, pesticides, plant health, and the environment.

#### 2.2.3. Implementation of Codex Standards

As noted above, good governance is also supported through the availability of internationally recognized standards that support AMR governance at the national level. FAO, together with WHO, is a parent organization of the Codex Alimentarius Commission, an intergovernmental body responsible for establishing food safety and quality standards that protect consumers and facilitate fair practices in the food trade. Codex has recently completed work in updating and developing new standards to support the management and containment of foodborne AMR [26]. Over a period of almost 5 years, Codex members and observers, together with input on the science from FAO and WHO, worked to develop an up-to-date triad of standards fit to address the AMR problems of today. This work was initiated in response to the global action plan on AMR. The standards are anchored in a One Health approach and provide a global reference, together with those of other standard-setting organizations such as the WOAH, on which to build national AMR stewardship initiatives on aspects ranging from monitoring of AMU to principles to guide the use of antimicrobials across the food chain.

During their development, the international standards adopted by the Codex Alimentarius Commission brought countries from around the world together to recognize and discuss the challenges posed by AMR in the food chain and the actions that need to be taken by all countries to manage and contain the problem. Getting agreement on all aspects of the new and revised Codex texts was challenging and required years of discussion and negotiation, further challenged with all interactions being moved to the virtual domain due to COVID. There were many bridges to build, and the importance of addressing an issue as significant as AMR in a One Health manner encouraged countries to build those bridges. This good will reared means there is now an opportunity to seek the engagement and support of countries to implement what was agreed in the Codex standards: developing surveillance programs, taking a risk-based approach, addressing AMR along the food chain, and adhering to agreed principles on the responsible and prudent use of antimicrobials to name but a few. The Republic of Korea is leading the way here, supporting FAO with a five-year project to implement the Codex standards on AMR. This project focuses in particular on developing monitoring and surveillance programs and implementing measures to minimize and contain foodborne AMR. Working with pilot countries in Asia and Latin America that are in different stages of their national AMR response planning and implementation, the project will enable FAO to gain practical experience in implementing these Codex standards, which can then be shared to help drive and facilitate their implementation elsewhere and support a somewhat harmonized approach to the management and containment of AMR along the food chain.

### 2.3. Practices

The establishment of an enabling environment for change through awareness raising, education, financing, monitoring, and appropriate governance contributes to the ability of individuals, organizations, and nations to address the problem of antimicrobial resistance. However, until such time that behaviors (i.e., agriculture management practices) change, there will be limited progress on reducing and mitigating the agricultural contributions of AMR.

#### 2.3.1. Reducing/Limiting Antimicrobial Use When Not Necessary

Fundamental to antimicrobial stewardship in the agrifood sectors is limiting the use of antimicrobials. Reducing the use of antimicrobials when they are not needed or when they are used improperly is the first step in antimicrobial use (e.g., wrong dose, route, or indication, for example, for viral infections). Stewardship may lead to the maintenance of the efficacy of the drugs as well as an economic savings for producers. Consultation of health professionals and access to appropriate diagnostics is critical for appropriate use. FAO has developed a number of tools and guidelines for raising animals (e.g., fish, beef and dairy cattle, swine, poultry, and bees) healthily with decreased use of antimicrobials.

#### 2.3.2. Good Production Practices

In addition to using antimicrobials responsibly, a large number of practices can be employed to reduce the future need for antimicrobials in plant and animal production.

In plant production, FAO has provided some suggestions on how to reduce the need for antimicrobials in plant production, specifically through the use of integrated pest management (IPM) practices [27]. Additional tools for plant producers are under development.

A wide range of animal production practices contributes to increasing animal wellbeing and making them more resistant and resilient to diseases, therefore minimizing the need to use antimicrobials. Some examples:Applying good husbandry practices while handling the animals, in the animal production establishments, and during animal transport;Improving animal welfare (e.g., ensuring good air and water supply quality, appropriate ventilation rates, and space allocation) during all phases, including production, transport, and slaughter;Using animals of locally adapted breeds that are more resistant to diseases and stress or animals bred for disease resistance (resistant animals will require a lower number of treatments with antimicrobials);Ensuring good hygiene, biosecurity measures, and general conditions on farms and during animal transport; these practices prevent the transmission of AMR;Applying rigorous disease control measures (e.g., vaccination), antibiotics are frequently used to treat bacterial infections that are secondary to many diseases that are preventable by vaccination. Preventing these diseases reduces the need for antimicrobials;Applying good practices for waste management.

In addition, functional animal nutrition to promote animal health is one of the most powerful tools available to decrease the need for antimicrobials in animal production. Nutrition affects the critical functions required for host defense and disease resistance. Animal nutrition strategies should therefore aim to support these host defense systems and reduce the risk of the presence in feed and water of potentially harmful substances, such as mycotoxins, antinutritional factors (e.g., lectins, and protease inhibitors) and pathogenic bacteria and other microbes.

General dietary measures to promote gastrointestinal tract health include the selective use of a combination of feed additives and feed ingredients to stabilize the intestinal microbiota and support mucosal barrier function.

Good nutrition allows the expression of the genetic potential of animals for different traits, including resistance to disease or stress, growth, milk or egg production, and reproductive functions. These depend on the availability of sufficient (preferably local) feed resources around the year, the genetic makeup of the animals, climatic and disease conditions, and husbandry practices.

Livestock nutrition programs are supported by diverse classes of feed additives, which have already been developed, marketed, and used in daily practice. Many of these functional additives are based on traditional fermentation techniques (prebiotics, probiotics, and synbiotics) and preservation technologies (organic acids). Other feed additives, such as phytochemicals, have their roots in traditional health practices and ethno-veterinary medicine. Evaluating locally available feed ingredients and traditional remedies based on herbal products abundantly available in the local environment should be integrated into the feeding strategy to reduce the need for the use of antimicrobials in livestock production.

Good nutrition also supports the critical functions required for a healthy gastrointestinal tract, host defense, and health. Various feeding practices can be used to reduce the presence of potentially harmful contaminants (e.g., pathogenic bacteria and natural toxins such as mycotoxins) and antinutritional factors in feed and water. Such practices include:Ensuring drinking water quality. The consumption of water of appropriate quality is a prerequisite for animal health. Regular control of the quality, supply and accessibility of water, and regular sanitation of water storage and delivery systems using disinfecting agents, are important measures to keep animals healthy. However, this could prove a challenge in regions with water shortages or high levels of water pollution;Ensuring feed safety and quality. Measures to ensure feed safety and quality include: minimizing the presence of microbiological, chemical, and physical hazards; ensuring appropriate levels of available protein, energy, and other nutrients and micronutrients to meet the requirements of the animal and ensure productivity, and ensuring appropriate physical characteristics such as particle size and pellet durability and hardness. Risk management in relation to the safety of feed and feed ingredients is an essential part of good feed production and manufacturing practices;Precision feeding. Knowledge of the nutritional requirements of species and breeds, and their specific needs at different life phases, has advanced feeding regimes, furthering sustainable production levels over the entire lifespan. Milestones in advancing feeding practices include the increasing availability and use of high-quality proteins, vitamins, chelated minerals, feed preservatives, and enzymes such as phytases, which all improve feed utilization. While these practices are proven to be effective at the producer level, their success partly depends on the safety and quality of feed and feed ingredients, which vary in nutrient and digestible energy content. In many countries, the availability of feed and feed ingredients of sufficient quality at every time of the year is an increasing concern. Agricultural practices, feed processing (mixing and pelleting), and the level of education of animal nutritionists and producers are key determinants of successful animal nutrition programs.

Feed additives. These are intentionally added ingredients not normally consumed as feed by themselves, whether or not they have nutritional value, which affects the characteristics of feed or livestock products. Diverse classes of feed additives have been developed, marketed, and used in livestock practices. The total market value was estimated at USD 38 billion in 2021 and was expected to reach USD 50 billion by 2026. These comprise prebiotics, probiotics, synbiotics, organic acids, and phytogenic compounds. A wide range of feed additives can be recommended to foster gastrointestinal and overall health, even under physiological or environmental stressful conditions such as weaning and regrouping, heat stress, undesirable antinutritional factors, and contaminants such as toxins. Such feed additives are promoted based on their effect on gut health, thereby improving feed utilization, the gut-associated immune system, and resilience to infectious diseases. While AGPs aim to stabilize the intestinal microbiota, a similar result can be achieved with non-antimicrobial compounds, which balance the microbiome and stimulate digestive enzymes and nutrient transport across a functional intestinal barrier. An improvement of intestinal health directly results in an improvement of the immune competence of an animal and hence overall resilience against infectious diseases. Improving gut health increases feed efficiency and, in turn, growth rate and productivity over the entire lifespan in all livestock species. Therefore, feed additives can not only replace AGP use in the improvement of gut health and immune competence but can also gradually reduce the need for antimicrobials for veterinary medical purposes. However, the efficacy and consistency of many feed additives can vary and are affected by feed composition, animal health and welfare, management practices, and the physical and social environment. FAO has already issued publications providing detailed information on relevant good practices, for example, the FAO/IFIF manual of Good Practices for the Feed Sector [28], the FAO/IDF Guide to Good Dairy Farming Practices [29], FAO Good Practices for Biosecurity in the Pig Sector [30], and continues working to promote them with a variety of stakeholders in the animal production sector.

### 2.4. Surveillance

Surveillance and research are essential to guide stakeholder decisions on how best to slow the emergence and spread of AMR for the good of food security and the health of humans, plants, animals, and the environment, globally. Strong surveillance and monitoring programs collect risk-based epidemiological and microbiological data on AMR, AMU, and antimicrobial residues relevant to each agriculture sub-sector and specific value chain that can be integrated with surveillance in humans and the environment. In general, surveillance for tracking antimicrobial resistance among bacteria and fungi recovered from human specimens is far more advanced than those for isolates from sick and healthy animals, from plants, food, and the environment, especially in low- and middle-income countries.

This information then allows for timely assessment of hazards to feed risk assessments to develop appropriate interventions and monitor their effectiveness over time for minimizing and containing AMR. Reliable data are needed on antimicrobial-resistant microorganisms—their distribution, AMR profiles, and prevalence—in addition to data on the extent of antimicrobial use (AMU) and antimicrobial residues along the food and feed chains, as well as through the various environments impacted by terrestrial (plant and animal) agriculture and aquaculture.

Harmonized data to estimate AMR levels in the food and agriculture sectors are still scarce. A recent study shows an increasing trend for antimicrobial resistance in common indicator pathogens (*Escherichia coli*, *Campylobacter* spp., nontyphoidal *Salmonella* spp., and *Staphylococcus aureus*) found in livestock. The study also shows AMR hotspots in several parts of the world [31].

The use of antimicrobials in human medicine is relatively well documented, but there are information gaps and inconsistencies in global estimates of volumes of antimicrobials used in animals. Previous studies have estimated that 73% of all antimicrobials sold globally are used in animals raised for food [32]. Studies of antimicrobial use trends and projections showed that in 2017, 93,309 tons of active ingredients was utilized for chicken, cattle, and pigs (which account for more than 90% of all food animals), and this amount was projected to increase by 11.5% by 2030 [33]. Pigs had the fastest projected growth in antimicrobial consumption (45%), while cattle had the smallest (22% of the global increase). Chickens contributed 33% to the total increase in antimicrobial use. Asia consumed the largest volume of antimicrobials with an expected growth of 10.3% by 2030. While Africa used lower quantities of antimicrobials in 2017 compared to other regions, it has the highest projected increase by 2030 (37%), but this amounts to just 6.1% of the world total in 2030. Aquaculture contributes 8% of animal protein intake to the human diet, and per capita consumption is increasing faster than for meat and dairy. It is estimated that it will use 5.7% of total antimicrobials by 2030, the highest use intensity per kilogram of biomass. The Asia – Pacific region represents the largest share (93.8%) of world consumption, with China alone contributing 57.9% of the total in 2017 [33]. Overuse and abuse of these drugs are observed as a replacement for good biosecurity practices in animal production. This contributes to the increased emergence and spread of antimicrobial resistance in pathogens, causing drug-resistant infections in animals and humans across the globe [34]. In parallel, the WOAH has published its Annual Report on Antimicrobial Agents Intended for Use in Animals [35]. Based on mostly sales and import data reported, it was estimated that a total of 69,455 tons of antimicrobial agents intended for use in animals were used in 2018. When analyzing trends in AMU from 2016 to 2018, the collected data, representing 65% of the global animal biomass, indicates an overall decrease of 27% in the mg/kg at the global level, moving from 120 mg/kg in 2016 to 88 mg/kg in 2018.

The AMU in plant production and protection has been less documented compared to the knowledge that we have regarding humans and animals. It seems to be comparatively small [36] but nevertheless should be further explored from a One Health perspective and to contribute to future risk management decisions.

#### 2.4.1. Assessing Laboratory and Surveillance Capacity

While AMR/AMU surveillance in humans, livestock, and food has developed faster in some countries than others, the inclusion of some sectors such as plant health, aquaculture, and the environment (e.g., contamination through animal waste, run-off) needs to be strengthened. Over the past few years, different initiatives worldwide have focused on and supported the generation of AMR surveillance data from the food and agriculture sectors, especially in LMICs. For this purpose, the FAO has developed and deployed a diagnostic instrument, the Assessment Tool for Laboratories and AMR Surveillance Systems (FAO-ATLASS), that assesses and defines targets to improve national AMR surveillance systems in the food and agriculture sectors. This tool has been used successfully used in more than 45 countries and 170 laboratories to date. More specifically, FAO-ATLASS is composed of two modules: the surveillance module and the laboratory module. Each module includes two standardized questionnaires that are completed by trained assessors in collaboration with the country representatives. Outputs of the assessment help countries identify limitations and gaps in their surveillance systems and assist them in the prioritization of investment and the development of roadmaps for surveillance strengthening.

One of the key elements critical to strengthening country capacities for surveillance and monitoring of AMR and AMU in food and agriculture is to ensure countries employ standardized approaches to collecting, analyzing, interpreting, and sharing data. However, often, as identified by ATLASS assessments, data are often not analyzed or used to inform decisions. Frequently reported reasons for this disconnect in data collection and use are the lack of adequate data management systems, unclear definition of roles and responsibilities in data reporting, and the dearth of trained personnel able to perform analysis and interpretation. FAO plans to continue supporting members in building and consolidating laboratory and surveillance capacity to generate, collect, and analyze high-quality data within national surveillance systems across all food and agriculture sectors.

#### 2.4.2. Data Storage, Analysis, and Interpretation

The FAO has long-standing experience developing and providing hosting data platforms to members for safe data storage and tools for data analysis and interpretation. FAO’s role as a neutral, impartial, independent, and specialized agency of the United Nations places the organization in the best position to establish a hosting data platform to support members for storing, analyzing, and using their own AMR data generated at a national level from the food and agriculture sectors.

FAO is committed to developing the building blocks that will catalyze national efforts to regularly generate and analyze reliable and comparable AMR data in food and agriculture and AMU data in plants and crops. With this purpose in mind, FAO started to develop a prototype for the International FAO Antimicrobial Resistance Monitoring (InFARM) IT platform in early 2022. FAO will be working with an initial set of countries to participate in developing and testing this data platform prototype during 2022. Countries will be involved in activities for pilot testing using their own data.

The development of InFARM will follow a progressive approach. The initial scope of the prototype of the InFARM IT platform will be to host AMR data in priority bacterial species of interest for public health, animal health, and indicator bacteria from animals and food, according to international standards and recommendations from Codex Alimentarius guidelines for integrated monitoring and surveillance of foodborne AMR and animal health codes of the World Organization for Animal Health (WOAH, founded as OIE). Global rollout of the InFARM IT platform and expansion to additional functionalities and surveillance priorities under FAO’s remit, such as data on the use of antimicrobial pesticides in plants, are planned upon completing the pilot phase, as baseline data and the capacity to monitor AMR and AMU in the plant health sector are lacking.

The InFARM IT platform will build momentum for enabling national surveillance, facilitating the process of data analysis, integration, and national reporting, and sustaining the monitoring of the country’s progress in actions against AMR in food and agriculture. This momentum is envisioned to further support the expansion of the InFARM platform as a wider system to support national surveillance activities such as formalization and establishment of national surveillance networks; capacity building of national focal points and experts on data generation and sharing; and adaptation of data management solutions and the interoperability with laboratory information management systems.

The objectives of the InFARM system and IT platform are:To support countries in collecting, analyzing, and using their AMR data from animals and food for national purposes. InFARM will support capacity building for global harmonization of AMR data generated by laboratory-based surveillance and will serve as a one-stop source of relevant contextual information on AMR and AMU surveillance programs and linked activities (e.g., national surveillance plans, reports of the application of FAO as-sessment tools);To support countries willing to publicly share AMR data from food and agriculture sectors for global surveillance as a public common good for international advocacy and action to tackle AMR. InFARM will include support for sharing data (with the possibility of protecting country identity by aggregation at regional and subregional levels) into the global tripartite integrated system for surveillance of AMR/AMU (TISSA). Public information on AMR prev-alence from InFARM will be the data source for outcome indicators included in the Monitoring & Evaluation (M&E) framework of the global action plan on AMR [37] (i.e., prevalence of resistant *E. coli* in animals) to measure the impact of actions to minimize and contain AMR in food and agriculture sectors at the global level.

## 3. Opportunities: Recent Global Efforts to Support AMS

### 3.1. Partnership Platform

Innovative and targeted solutions focused on system strengthening and sustainable development should be put forward, while the cooperation of the whole of society and all sectors is paramount to ensure antimicrobials remain effective.

A multi-stakeholder partnership approach can contribute in a unique way to help build this global coalition of the willing and catalyze knowledge and evidence share, political ownership, innovative financing models and incentives, and targeted actions at all levels to combat AMR through the One Health approach.

Pursuant to the UN Interagency Coordination Group on AMR (IACG) report recommendation to ‘establish a constituency-based partnership platform to develop and work towards a shared global vision, goals and coordinated action on AMR’ and given the need to leverage local and sectoral knowledge on AMR and engage a wide variety of stakeholders from the public and private sectors, to generate and sustain systemic change, the quadripartite (FAO, UNEP, WOAH, and WHO) put in place a multi-stakeholder partnership platform on AMR as part of the AMR global governance along with the GLG and IPEA.

The platform aims to be an inclusive, international, and multi-stakeholder forum, which brings together relevant actors across the human, animal, plant, and environment interface to assist in preserving antimicrobials as lifesaving medicines and ensuring their responsible use under a One Health approach. It promotes a shared global vision, builds consensus, and takes targeted action to address the growing global threat of AMR through a coordinated, multisectoral, inclusive One Health approach to contribute to the achievement of the 2030 Agenda for Sustainable Development, universal health coverage, and strengthening of health systems for future pandemic prevention and preparedness.

Given that the multisectoral work on AMR requires engagement from health, agriculture, environmental, pharmaceutical, trade, finance, and other sectors, the multi-stakeholder dimension of the platform enables deliberate coordination and collaboration within and between these sectors and stakeholders. It allows stakeholders such as governments, UN organizations and their specialized bodies, regional and intergovernmental organizations, civil society, research and academia institutions, private sector companies, philanthropic organizations, and multilateral organizations to make their voices heard, share their expertise, knowledge, and lessons learned, align positions, and catalyze actions to tackle AMR at community, national, regional, and global level.

The expected outcomes of the platform include:Improved dialogue, coordination, and collective public engagement on AMR targeting sectors, disciplines, and levels within countries and territories, as well as with quadripartite, AMR governance structures (GLG, IPEA, One Health High-Level Expert Panel (OHHLEP)) and other relevant stakeholders such as UN agencies, multilateral organizations, resource partners;Better common understanding of the AMR challenges and opportunities at community, country, regional and global levels for efficient and evidence-based decision making and action;Sustainable and effective actions to tackle AMR through One Health operationalization at community, country, regional and global levels;Emerging innovative public-private partnerships and alliances contributing to building the economic case of AMR, promoting the global public health research agenda, etc.;Heightened awareness, political uptake, and stakeholder ownership to tackle AMR through a One Health approach.

Currently, there are many ongoing AMR activities across the globe, yet as the COVID-19 pandemic highlighted, siloed approaches and initiatives focusing on specific aspects of health without looking through holistic One Health lenses may jeopardize global AMR containment efforts. The platform can potentially be an instrument to break these silos at all levels by operationalizing the One Health approach for AMR and creating an enabling environment for strengthened coordination, dialogue, and joint efforts to advance the delivery of the GAP, NAPs, and 2030 agenda and its SDGs. The platform’s geographical coverage is global, but a special focus is on LMICs and underrepresented stakeholders.

### 3.2. Reduce the Need for Antimicrobials in Agrifood Systems

Agrifood systems are facing all-time challenges as the global food demand is rapidly rising, estimated to increase by 35–56% between 2010 and 2050 [38]. This results in intensifying production, which drives an overall increase in the use of antimicrobials within agrifood systems. Indeed, in order to effectively fight AMR under a One Health approach while ensuring sustainable food production, it is critical that the necessary transformation of current agrifood systems into sustainable production includes a reduction in the need to use antimicrobials.

Antimicrobials are a precious discovery; their use has allowed saving millions of lives and providing food at levels that were impossible before. However, we have reached a point where its use in our current agrifood production systems, while providing positive outcomes, is also compromising that same use. It is for that reason that the call to reduce the need for antimicrobials is becoming clearer than ever. The Global Leaders Group on AMR in August 2021 released the following statement: “Applying a One Health approach, there is a critical need to transform food systems to optimize animal, plant, and environmental health, ensure responsible and sustainable antimicrobial use, and most importantly, reduce the need to use antimicrobials and promote innovation for evidence-based and sustainable alternatives” [39].

The FAO Committee on Agriculture (COAG) subcommittee on livestock has recommended requesting FAO collect scientific evidence on alternative feeding practices to replace the use of medically important antimicrobials used as growth promoters (AGPs), their effectiveness and safety, and to develop an inventory of these practices and disseminate related knowledge. The subcommittee has also recommended sharing successful experiences and good practices, including traditional knowledge, to support members in reducing the need for antimicrobials.

A “Reduce the need” initiative would, therefore, greatly benefit from an FAO’s unique comparative advantage as a multisectoral and multidisciplinary organization with solid internal institutional architecture and has been working actively in many countries around the world supporting national One Health platforms.

## 4. Conclusions

The control of the currently alarming levels of AMR is the realistic goal that we aim to achieve as a global community, as AMR will always exist, being a natural phenomenon exacerbated by the overuse of antimicrobials in humans, animals, and plants. The concept of stewardship is challenging, as it can be perceived as abstract. In any case, emphasizing the idea that antimicrobials are common goods that need to be preserved by all of us is essential, as it will ultimately lead to a healthier and more sustainable future. In times when food security cannot be taken for granted, the role of the FAO as a global organization is highlighted in this paper, and its multiple initiatives, projects, studies, and campaigns are presented.

## Figures and Tables

**Figure 1 microorganisms-10-01599-f001:**
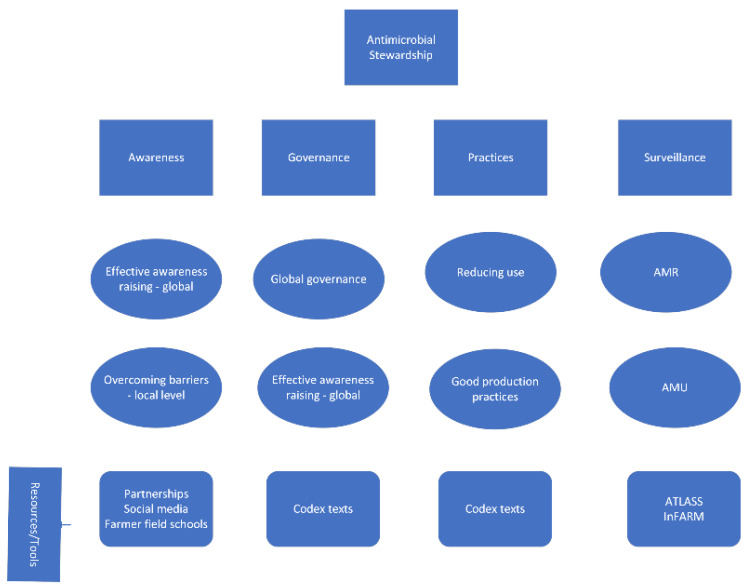
Key supporting pillars for antimicrobial stewardship.

## Data Availability

Not applicable.

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
