# Peer review of "Achieving Antimicrobial Stewardship on the Global Scale: Challenges and Opportunities"

_microorganisms, 2022, doi:10.3390/microorganisms10081599_

Round 1

Reviewer 1 Report

Time article to approach antimicrobial resistance and stewardship from a one health and a policy approach. 

Some comments and revisions:

1. In the introduction, the authors tried describe AMR in layman's terms and tried to illustrate that focusing on anti-bacterial agents is a narrow view. This is laudable, but it is not well phrased, and the meaning is lost (this is in the first 2 lines of page 2. An alternative recommendation is as follow.

"in practical terms, the view of AMR is often simplistic and narrow, referring only to the failure of antibiotics to treat human bacterial infections" 

2. In the 2nd last paragraph on Page 2, the authors used an abbreviation "Antimicrobial use (AMU) stewardship - but antimicrobial stewardship is an established phrase and this would be preferred instead - e.g. antimicrobial stewardship (AMS)

3. In general the structure is well through through, but the authors would need to quantify their statements / opinion more objectively. 

(a) For the awareness raising materials - linking the web links alone is not useful. It does not describe the content. The authors could describe this in text in a little bit more detail, and reference the relevant articles. The list of web links was not useful, and did not add to the text. 

(b) In regard to the webinars and workshops conducted on AMR and AMU in the plant health sector - (page 4 - second bullet point) - what is the scope of the material? This could be better defined. AMR and AMU can be very specialised subject matters, and the authors talked abt improving farmer's knowledge (I was wondering how this is measured /quantified in real life. Perhaps the first 2 bullet points on page 4 has to be reorganised slightly to convey the right message.

(c) On the subject of Surveillance on Page 10 of 15, can the authors describe the global picture of AMU / AMR in the agricultural industry?

- In regard to antimicrobial use

- in regard to surveillance for antimicrobial resistancee 

While the data for AMR / AMU may be scare, there is still some data, could the authors describe this in relation to the global situation? 

IT would be good also to discuss A proposed and practical framework for surveillance (page 10)

4. Can the authors clarify what do you mean by this statement on Page 10?

"research and data as are impact assessments of the use of antimicrobials in plant production systems on human, animal and environmental health. (please revise).

5. Revision of the language throughout the text can be improved upon.

E.g. in the abstract, certain phrasing makes the text difficult to read and meaning difficult to grasp.

E.g. "Consequently, antimicrobial use stewardship needs to be increased, if one is to react control of the AMR phenomenon." This sentence needs clarity.

Did the authors mean - It is imperative for antimicrobial stewardship to be more widely implemented to achieve better control of the AMR problem.

"Through the application of various tools and support, in particular in low-and middle-income countries, the specific needs and barriers to adopt and implement antimicrobial stewardship practices in agrifood systems are being identified. As a global community, it is essential that we move beyond needs and problems assessment to the solutions implementation. Thus, FAO also provides support for nations to improve antimicrobial stewardship through programs to strengthen the governance, awareness, surveillance and best practices related to antimicrobial resistance in agrifood systems." (text in italics just does not flow.

Suggestion:

"Through the application of various tools and support, in particular in low-and middle-income countries, the specific needs to address and barriers to overcome in order to implement antimicrobial stewardship practices in agrifood systems are being identified. As a global community, it is essential that we now move beyond discussing the AMR problem and focus on implementing solutions. Thus, FAO also provides support for nations to improve antimicrobial stewardship through programs to strengthen the governance, increase awareness, develop and enhance AMR surveillance and implement best practices related to antimicrobial resistance in agrifood systems." (recommendations in bold) 

Please check the grammar throughout the text, there are quite a few grammatical errors.

Author Response

Roma, Italy

July 25, 2022

Dear Editorial Board,

Thank you very much for attention to our manuscript “Achieving antimicrobial stewardship on the global scale: challenges and opportunities”, and the careful and constructive revisions by the two reviewers.

We have revised our manuscript accordingly – please find it attached.

Specifically, on the points raised by the reviewers:

Reviewer 1:

Time article to approach antimicrobial resistance and stewardship from a one health and a policy approach. 

Thank you for the positive feedback.

Some comments and revisions:

  1. In the introduction, the authors tried describe AMR in layman's terms and tried to illustrate that focusing on anti-bacterial agents is a narrow view. This is laudable, but it is not well phrased, and the meaning is lost (this is in the first 2 lines of page 2. An alternative recommendation is as follow.

"in practical terms, the view of AMR is often simplistic and narrow, referring only to the failure of antibiotics to treat human bacterial infections" 

We have reviewed the definition of “antimicrobial”, stating now the one that was recently adopted by the FAO, and included the edited sentence has suggested by the reviewer.   

  1. In the 2nd last paragraph on Page 2, the authors used an abbreviation "Antimicrobial use (AMU) stewardship - but antimicrobial stewardship is an established phrase and this would be preferred instead - e.g. antimicrobial stewardship (AMS)

AMS is now used throughout the manuscript.

  1. In general the structure is well through through, but the authors would need to quantify their statements / opinion more objectively. 

(a) For the awareness raising materials - linking the web links alone is not useful. It does not describe the content. The authors could describe this in text in a little bit more detail, and reference the relevant articles. The list of web links was not useful, and did not add to the text. 

(b) In regard to the webinars and workshops conducted on AMR and AMU in the plant health sector - (page 4 - second bullet point) - what is the scope of the material? This could be better defined. AMR and AMU can be very specialised subject matters, and the authors talked abt improving farmer's knowledge (I was wondering how this is measured /quantified in real life. Perhaps the first 2 bullet points on page 4 has to be reorganised slightly to convey the right message.

We actually deleted the reference to these materials, as, when reviewing the manuscript, we felt that they were not bringing much added value to the manuscript.

(c) On the subject of Surveillance on Page 10 of 15, can the authors describe the global picture of AMU / AMR in the agricultural industry?

- In regard to antimicrobial use

- in regard to surveillance for antimicrobial resistancee 

While the data for AMR / AMU may be scare, there is still some data, could the authors describe this in relation to the global situation? 

IT would be good also to discuss A proposed and practical framework for surveillance (page 10)

We added a fair amount of new content (text and references) in this section, addressing the reviewer comments accordingly.

  1. Can the authors clarify what do you mean by this statement on Page 10?

"research and data as are impact assessments of the use of antimicrobials in plant production systems on human, animal and environmental health. (please revise).

This sentence was indeed not clear, so we actually deleted it, and hopefully clarified the idea, in the added new text.

  1. Revision of the language throughout the text can be improved upon.

E.g. in the abstract, certain phrasing makes the text difficult to read and meaning difficult to grasp.

E.g. "Consequently, antimicrobial use stewardship needs to be increased, if one is to react control of the AMR phenomenon." This sentence needs clarity.

Did the authors mean - It is imperative for antimicrobial stewardship to be more widely implemented to achieve better control of the AMR problem.

"Through the application of various tools and support, in particular in low-and middle-income countries, the specific needs and barriers to adopt and implement antimicrobial stewardship practices in agrifood systems are being identified. As a global community, it is essential that we move beyond needs and problems assessment to the solutions implementation. Thus, FAO also provides support for nations to improve antimicrobial stewardship through programs to strengthen the governance, awareness, surveillance and best practices related to antimicrobial resistance in agrifood systems." (text in italics just does not flow.

Suggestion:

"Through the application of various tools and support, in particular in low-and middle-income countries, the specific needs to address and barriers to overcome in order to implement antimicrobial stewardship practices in agrifood systems are being identified. As a global community, it is essential that we now move beyond discussing the AMR problem and focus on implementing solutions. Thus, FAO also provides support for nations to improve antimicrobial stewardship through programs to strengthen the governance, increase awareness, develop and enhance AMR surveillance and implement best practices related to antimicrobial resistance in agrifood systems." (recommendations in bold) 

Suggestions were included, as proposed.

Please check the grammar throughout the text, there are quite a few grammatical errors

All the text was revised, to correct, for example, the mentioned grammatical errors.

Thankful in advance for your attention, we look forward to hearing back from you.

Best regards,

Jorge Pinto Ferreira 

Reviewer 2 Report

Dear authors,

First of all, I would like to highlight the effort to write this review, however from my point of view I think that there are hard work forward publishing it.

The grammar and english style could be improved in order to make easier to follow the text.

f. example: To begin understanding....

Also notes like "An anciente Chinese proverb state" is out of the range...

And the most important for me, what is the point of this review? what do the author want to transmit or review...? It is not clear for me. Besides, why only 3 chapters with multiple sections and subchapters?Maybe, it will be a good chance to make more chapters and be more schematic for each one (too text to read now!)

So, briefly and from the beging

- Introduction: please, coulf the author be more creative definit the antimicrobials. the definition looks to copy and paste from wikipedia.

-Maybe I will remove the 3 first paragraph or join in one. In addition because they do not have anu references.........

- From my point , could be good to comment about ONE HEALTH concept

- The URLs please, explain them and move the links to references, but not show them in the main text

-check the concept of AMR, because maybe ONE HEALTH  is more appropiate.

And,  please re-schedule the chapters and rewrite according to .

NOTE: this paper, just in case, could help to clarify the schedule to follow

https://doi.org/10.1016/j.onehlt.2021.100339

Thanks

Author Response

Roma, Italy

July 25, 2022

Dear Editorial Board,

Thank you very much for attention to our manuscript “Achieving antimicrobial stewardship on the global scale: challenges and opportunities”, and the careful and constructive revisions by the two reviewers.

We have revised our manuscript accordingly – please find it attached.

Specifically, on the points raised by the reviewers:

Reviewer 2

First of all, I would like to highlight the effort to write this review, however from my point of view I think that there are hard work forward publishing it.

The grammar and english style could be improved in order to make easier to follow the text.

  1. example: To begin understanding....

All the text was revised, to improve the grammar and English style.

Also notes like "An anciente Chinese proverb state" is out of the range...

This sentence was deleted.

And the most important for me, what is the point of this review? what do the author want to transmit or review...? It is not clear for me.

The last sentence of the introduction was edited accordingly, to provide a better clarification of the purpose of the paper.  

Besides, why only 3 chapters with multiple sections and subchapters?Maybe, it will be a good chance to make more chapters and be more schematic for each one (too text to read now!)

We have reviewed the structure of the paper, having now more chapters, sub chapters and a schematic figure.

So, briefly and from the beginning

- Introduction: please, coulf the author be more creative definit the antimicrobials. the definition looks to copy and paste from wikipedia.

We have reviewed the definition of antimicrobial, and have now included the one recently adopted by FAO.

-Maybe I will remove the 3 first paragraph or join in one. In addition because they do not have anu references.........

The three first paragraphs were joined into one.

- From my point , could be good to comment about ONE HEALTH concept

We added a definition of One Health, and commented on its application.

- The URLs please, explain them and move the links to references, but not show them in the main text

We actually deleted them, as we felt they were not of much added value to the publication.

-check the concept of AMR, because maybe ONE HEALTH  is more appropiate.

One Health concept is now better clarified in the publication.

And,  please re-schedule the chapters and rewrite according to .

NOTE: this paper, just in case, could help to clarify the schedule to follow

https://doi.org/10.1016/j.onehlt.2021.100339

The paper structure was reviewed, having now more chapters, sub chapters, and a schematic figure.

Thankful in advance for your attention, we look forward to hearing back from you.

Best regards,

Jorge Pinto Ferreira

Round 2

Reviewer 1 Report

Thank you for the edits. 

The paper is clearer. No comments.